# Effects of Ruxolitinib and Calcitriol Combination Treatment on Various Molecular Subtypes of Breast Cancer

**DOI:** 10.3390/ijms23052535

**Published:** 2022-02-25

**Authors:** Jean Schneider, Ye Won Jeon, Young Jin Suh, Seung Taek Lim

**Affiliations:** 1College of Natural Sciences, University of Texas at Austin, Austin, TX 78705, USA; jsschneider4055@gmail.com; 2Division of Breast and Thyroid Surgical Oncology, Department of Surgery, St. Vincent’s Hospital, College of Medicine, The Catholic University of Korea, Seoul 16247, Korea; yeswon80@naver.com (Y.W.J.); yjsuh@catholic.ac.kr (Y.J.S.)

**Keywords:** ruxolitinib, calcitriol, combination, breast cancer, molecular subtype

## Abstract

The anticancer effects of ruxolitinib and calcitriol against breast cancer were reported previously. However, the effect of ruxolitinib and calcitriol combination treatment on various molecular subtypes of breast cancer remains unexplored. In this study, we used MCF-7, SKBR3, and MDA-MB-468 cells to investigate the effect of ruxolitinib and calcitriol combination treatment on cell proliferation, apoptosis, cell cycle, and cell signaling markers, in vitro and in vivo. Our results revealed the synergistic anticancer effect of ruxolitinib and calcitriol combination treatment in SKBR3 and MDA-MB-468 cells, but not in MCF-7 cells in vitro, via cell proliferation inhibition, apoptosis induction, cell cycle arrest, and the alteration of cell signaling protein expression, including cell cycle-related (cyclin D1, CDK1, CDK4, p21, and p27), apoptosis-related (c-caspase and c-PARP), and cell proliferation-related (c-Myc, p-p53, and p-JAK2) proteins. Furthermore, in the MDA-MB-468 xenograft mouse model, we demonstrated the synergistic antitumor effect of ruxolitinib and calcitriol combination treatment, including the alteration of c-PARP, cyclin D1, and c-Myc expression, without significant drug toxicity. The combination exhibited a synergistic effect in HER2-enriched and triple-negative breast cancer subtypes. In conclusion, our results suggest different effects of the combination treatment of ruxolitinib and calcitriol depending on the molecular subtype of breast cancer.

## 1. Introduction

Breast cancer is the most frequently diagnosed cancer and was the most common cause of cancer-specific mortality among women worldwide in 2020 [1]. It is estimated that in the United States, approximately 281,550 cases and 43,600 deaths will occur due to breast cancer in 2021 [2]. Despite recent advances in the therapeutic strategies for breast cancer, including surgical techniques and chemotherapy regimens, the prognosis in patients with breast cancer remains unsatisfactory [3]. One of the reasons that worsen a patient’s prognosis is the resistance to anticancer agents [4], which inevitably necessitates a novel therapeutic strategy, including the combination of anticancer agents that were previously used as a single agent. The determination of relevant combinations of anticancer drugs allows for a synergistic treatment effect compared with monotherapy [5].

Ruxolitinib is an oral selective inhibitor of Janus kinase (JAK) 1 and JAK2 [6] and was initially approved for use in the treatment of myelofibrosis and polycythemia vera [7,8]. Subsequently, the anticancer effect of ruxolitinib was investigated in various solid tumors, including non-small cell lung, bladder, hepatocellular, and breast cancer [9,10,11,12]. Despite the unsatisfactory results of ruxolitinib monotherapy for the treatment of breast cancer in previous clinical trials [13,14], many studies investigating the therapeutic effect of ruxolitinib in combination with other drugs are reported or are still ongoing [15,16].

Calcitriol, the active form of vitamin D (1,25-dihydroxy vitamin D_3_), plays an important role in calcium homeostasis through its actions in the intestine, kidneys, and bones [17]. In addition to its role in calcium homeostasis, several studies showed the potential use of calcitriol in the treatment of various cancers, including breast cancer [18,19,20]. Additionally, previous studies reported the synergistic effect of calcitriol and other anticancer agents’ combination treatment in breast cancer, in vitro and in vivo [21,22].

Breast cancer can be categorized into four distinct molecular subtypes, namely luminal A, luminal B, HER2-enriched, and triple negative (TN) subtypes, according to the expression status of hormone receptors (HR), including estrogen receptor (ER), progesterone receptor (PR), and human epidermal growth factor receptor 2 (HER2) [23,24]. In our previous study, we reported the synergistic treatment effect of ruxolitinib and calcitriol in luminal B subtype MCF7-HER18 breast cancer cells in vitro [25]. However, for the application of our previous results in various clinical situations, the combined anticancer effect of ruxolitinib and calcitriol should be investigated in other breast cancer molecular subtypes because the treatment response to anticancer agents can differ depending on the molecular subtype of breast cancer in a patient [26]. Therefore, in this study, we investigated the anticancer effects of ruxolitinib and calcitriol combination therapy and their molecular mechanisms in various molecular subtypes in vitro and in vivo.

## 2. Results

### 2.1. Ruxolitinib and Calcitriol Inhibit the Proliferation of MCF-7, SKBR3, and MDA-MB-468 Cells as a Single Agent In Vitro

The antiproliferative effect of ruxolitinib and calcitriol as single agents on MCF-7, SKBR3, and MDA-MB-468 cells was evaluated using the BrdU assay. In the BrdU assay, ruxolitinib significantly reduced cell proliferation in MCF-7 (20 μM: 79.39 ± 7.72%, 30 μM: 51.11 ± 2.58%), SKBR3 (10 μM: 83.48 ± 6.16%, 12 μM: 49.77 ± 3.96%, 20 μM: 23.68 ± 1.21%), and MDA-MB-468 cells (10 μM: 57.05 ± 3.06%, 20 μM: 19.53 ± 1.72%, 30 μM: 18.57 ± 2.71%) in comparison with respective controls (Figure 1A). The IC_50_ values of the ruxolitinib were 30.42 μM in MCF-7, 13.94 μM in SKBR3, and 10.87 μM in MDA-MB-468 cells. Additionally, calcitriol significantly reduced cell proliferation in MCF-7 (10 μM: 77.92 ± 4.21%, 15 μM: 69.21 ± 6.36%, 20 μM: 55.91 ± 2.43%), SKBR3 (100 nM: 83.48 ± 4.19%, 1 μM: 45.01 ± 1.61%, 10 μM: 32.28 ± 2.33%), and MDA-MB-468 cells (15 μM: 84.09 ± 8.85%, 20 μM: 75.66 ± 9.79%) compared with that in respective controls (Figure 1B). The IC_50_ values of the Calcitriol were 24.49 μM in MCF-7, 2.01 μM in SKBR3, and 135.42 μM in MDA-MB-468 cells.

### 2.2. Combination Treatment with Ruxolitinib and Calcitriol Synergistically Inhibits the Proliferation of SKBR3 and MDA-MB-468 Cells In Vitro

Considering the IC_50_ values of ruxolitinib, combination drug analysis was performed at a fixed concentration of ruxolitinib and various concentrations of calcitriol in MCF-7, SKBR3, and MDA-MB-468 cells (Figure 1C). In MCF-7 cells, a combination of ruxolitinib at 30 μM with various concentrations of calcitriol (10 μM, 15 μM, and 20 μM) did not significantly increase the antiproliferative effects (ruxolitinib at 30 μM and calcitriol at 10 μM: 51.62 ± 3.38%, ruxolitinib at 30 μM and calcitriol at 15 μM: 50.9 ± 2.63%, ruxolitinib at 30 μM and calcitriol at 20 μM: 50.31 ± 2.86%) when compared with ruxolitinib-only treatment at 30 μM (49.33 ± 6.66%). However, in SKBR3 cells, the combination of ruxolitinib at 12 μM and various concentrations of calcitriol (10 nM, 100 nM, and 1 μM) significantly increased the antiproliferative effect (ruxolitinib at 12 μM and calcitriol at 10 nM: 42.46 ± 1.99%, ruxolitinib at 12 μM and calcitriol at 100 nM: 36.73 ± 1.27%, ruxolitinib at 12 μM and calcitriol 1 μM: 33.15 ± 0.71%) when compared to that with ruxolitinib-only treatment at 12 μM (48.18 ± 3.31%). Similarly, in MDA-MB-468 cells, a combination of ruxolitinib at 10 μM and various concentrations of calcitriol (10 μM, 15 μM, and 20 μM) significantly increased the antiproliferative effects (ruxolitinib at 10 μM and calcitriol at 10 μM: 34.1 ± 1.06%, ruxolitinib at 10 μM and calcitriol at 15 μM: 26.97 ± 1.07%, ruxolitinib at 10 μM and calcitriol at 20 μM: 22.59% ± 1.99%) compared to those with ruxolitinib-only treatment at 10 μM (63.76 ± 4.38%).

Subsequently, the combination index (CI) at each drug combination concentration was calculated to determine the drug synergism, as shown in Table 1 and Figure 1D. In MCF-7 cells, CI at all combinations showed an antagonistic effect between ruxolitinib and calcitriol (CI 1.433–1.814). However, in SKBR3 and MDA-MB-468 cells, CI at all combinations showed a synergistic effect between ruxolitinib and calcitriol (SKBR3: CI 0.759–0.836, MDA-MB-468: CI 0.676–0.787). These results indicate that the combination treatment with ruxolitinib and calcitriol had a synergistic antiproliferative effect in SKBR3 and MDA-MB-468 cells but not in MCF-7 cells.

### 2.3. Combination Treatment with Ruxolitinib and Calcitriol Synergistically Induces Apoptosis in SKBR3 and MDA-MB-468 Cells In Vitro

In MCF-7 cells, although the combination treatment with ruxolitinib 30 μM and calcitriol 20 μM significantly increased apoptotic cell percentage (20.33 ± 2.30%) compared to that with the control (6.13 ± 1.89%) or calcitriol 20 μM (6.79 ± 1.23%) treatments, the combination treatment significantly decreased the percentage of apoptotic cells in comparison with ruxolitinib 30 μM treatment (30.65 ± 2.40%) (Figure 2A). However, in SKBR3 cells, the percentage of apoptotic cells significantly increased with the ruxolitinib 12 μM and calcitriol 100 nM combination treatment (61.83 ± 6.05%) compared to that with the control (1.95 ± 0.78%), ruxolitinib 12 μM (40.66 ± 7.42%), and calcitriol 100 nM (8.55 ± 3.02%) treatments (Figure 2B). Furthermore, in MDA-MB-468 cells, the percentage of apoptotic cells significantly increased with the ruxolitinib 10 μM and calcitriol 20 μM combination treatment (67.16 ± 5.00%) compared to that with the control (7.02 ± 1.36%), ruxolitinib 10 μM (48.9 ± 5.58%), and calcitriol 20 μM (10.31 ± 3.11%) treatments (Figure 2C). These results suggest a synergistic pro-apoptotic effect of ruxolitinib and calcitriol combination treatment in SKBR3 and MDA-MB-468 cells.

### 2.4. Ruxolitinib and Calcitriol as a Single Agent and Combination Treatment Induced Cell Cycle Arrest in MCF-7, SKBR3, and MDA-MB-468 Cells In Vitro

In MCF-7 cells (Figure 3A), unlike in the control (G0/G1 76.37 ± 2.83%, S 11.45 ± 2.46%, G2/M 10.09 ± 3.45%, Sub G1 2.08 ± 1.89%), treatment with ruxolitinib 30 μM significantly decreased the percentage of cells in the G0/G1 phase and increased the percentage cells in the G2/M phase (G0/G1 57.51 ± 3.4%, S 6.56 ± 2.08%, G2/M 32.54 ± 3.66%, Sub G1 3.38 ± 1.97). Treatment with calcitriol 20 μM significantly decreased the percentage of cells in the S phase and increased the percentage of cells in the G0/G1 phase (G0/G1 84.69 ± 4.26%, S 6.4 ± 1.46%, G2/M 6.01 ± 3.57%, Sub G1 2.86 ± 2.31%) compared to those in the control group. Furthermore, ruxolitinib 30 μM and calcitriol 20 μM combination treatment significantly decreased the percentage of cells in the G0/G1 phase and increased the percentage of cells in the G2/M phase (G0/G1 56.27 ± 4.88%, S 7.01 ± 1.88%, G2/M 33.79 ± 4.34%, Sub G1 2.91 ± 2.47%) compared to those in the control group.

In SKBR3 cells (Figure 3B), in comparison to the control (G0/G1 76.53 ± 2.4%, S 14.79 ± 2.39%, G2/M 6.47 ± 3.13%, Sub G1 2.19 ± 2.02%), treatment with ruxolitinib 12 μM significantly decreased the percentage of cells in the G0/G1 phase and increased the percentage of cells in the G2/M phase (G0/G1 69.69 ± 1.73%, S 14.08 ± 2.9%, G2/M 12.72 ± 1.61%, Sub G1 3.33 ± 2.72%). When treated with calcitriol 0.1 μM, there was a significant decrease in the percentage of cells in the S phase and an increase in the percentage of cells in the G0/G1 phase (G0/G 89.99 ± 3.26%, S 4.3% ± 1.15%, G2/M 4.2 ± 2.85%, Sub G1 1.5 ± 1.57%) in comparison to those in the control group. Additionally, ruxolitinib 12 μM and calcitriol 0.1 μM combination treatment caused a significant decrease in the percentage of cells in the S phase, which was accompanied by a non-significant increase in the percentage of cells in the G0/G1 and G2/M phases (G0/G1 78.96 ± 3.02%, S 7.53 ± 2.02%, G2/M 10.98 ± 2.46%, Sub G1 2.52 ± 3.02%) compared to those in the control group.

In MDA-MB-468 cells (Figure 3C), in comparison to the control (G0/G1 59.80 ± 5.15%, S 15.12 ± 2.31%, G2/M 17.23 ± 5.28%, Sub G1 7.83 ± 2.11%), treatment with ruxolitinib 10 μM significantly decreased the percentage of cells in the G0/G1 phase and increased the percentage of cells in the G2/M phase (G0/G1 48.05 ± 2.67%, S 12.57 ± 2.02%, G2/M 35.1 ± 2.73%, Sub G1 4.26 ± 1.97%). When treated with calcitriol 20 μM, a significant decrease in the percentage of cells in the S phase and an increase in the percentage of cells in the G0/G1 phase (G0/G1 176.18 ± 3.84%, S 6.16 ± 2.56%, G2/M 14 ± 3.48%, Sub G1 3.66 ± 3.06%) was observed. Furthermore, ruxolitinib 10 μM and calcitriol 20 μM combination treatment caused a significant decrease in the percentage of cells in the S phase and an increase in the percentage of cells in the G2/M phase (G0/G1 57.05 ± 3.37%, S 5.22 ± 2.52%, G2/M 32.86 ± 5.02%, Sub G1 4.86 ± 0.88%) compared to those in the control group. These results suggest that while the G2/M arrest effect of ruxolitinib and G0/G1 arrest effect of calcitriol were consistent in all the cell lines tested, the combined effect of ruxolitinib and calcitriol in the cell cycle was different among cell lines depending on their molecular subtypes.

### 2.5. Combination Treatment with Ruxolitinib and Calcitriol Synergistically Affects Cell Proliferation, Cell Cycle, and Apoptosis-Related Proteins in SKBR3 and MDA-MB-468 Cells In Vitro

In MCF-7 cells (Figure 4A), ruxolitinib 30 μM treatment significantly downregulated CDK1, p27, c-caspase 3, and p-JAK2 expression and upregulated c-PARP expression in comparison to that in the control. In MCF-7 cells treated with calcitriol 20 μM, significant downregulation of c-Myc, CDK1, and p27 expression was observed in comparison to that in the control group. Furthermore, when treated with the ruxolitinib 30 μM and calcitriol 20 μM combination, MCF-7 cells exhibited significant downregulation of p-p53 and upregulation of Bcl-2 expression compared to that observed with the control, ruxolitinib 30 μM treatment, and calcitriol 20 μM treatments.

In SKBR3 cells (Figure 4B), ruxolitinib 12 μM treatment significantly downregulated c-Myc, CDK1, and CDK4 expression and upregulated Bcl-2, c-PARP, and p-p53 expression compared to that in the control. Treatment with calcitriol 0.1 μM significantly downregulated c-Myc, cyclin B1, cyclin D1, CDK1, CDK4, and CDK6 expression and upregulated p21, p27, and Bcl-2 expression in SKBR3 cells. Furthermore, upon ruxolitinib 12 μM and calcitriol 0.1 μM combination treatment, SKBR3 cells showed significant downregulation of c-Myc, cyclin D1, CDK1, and p-JAK2 and upregulation of p21, p27, BAD, c-caspase 3 and p-p53 expression compared to that observed with control, ruxolitinib 12 μM treatment, and calcitriol 0.1 μM treatments.

In MDA-MB-468 cells (Figure 4C), ruxolitinib 10 μM treatment significantly downregulated cyclin D1, CDK4, CDK6, and BAD expression and upregulated p21, Bcl-xL, c-caspase 3, and c-PARP expression compared to that in the control. Calcitriol 20 μM treatment of MDA-MB-468 cells caused a significant downregulation of c-Myc, CDK4, CDK6, and p27 expression in comparison to that in the control. Furthermore, treatment of MDA-MB-468 cells with ruxolitinib 10 μM and calcitriol 20 μM resulted in significant downregulation of c-Myc, cyclin D1, CDK1, CDK4, and p-JAK2 expression and upregulation of p27 and c-PARP expression compared to that observed with control, ruxolitinib 10 μM treatment, and calcitriol 20 μM treatment.

These results suggest a synergistic anticancer effect of ruxolitinib and calcitriol combination treatment in SKBR3 and MDA-MB-468 cells by altering the expression of cell signaling proteins, including cell cycle-related (cyclin D1, CDK1, CDK4, p21, p27), apoptosis-related (c-caspase, c-PARP), and cell proliferation-related (c-Myc, p-p53, p-JAK2) proteins. Furthermore, these results suggest that ruxolitinib and calcitriol, as single agents or combination treatment, were linked to various cell signaling mechanisms in MCF-7, SKBR3, and MDA-MB-468 cells.

### 2.6. Combination Treatment with Ruxolitinib and Calcitriol Has a Synergistic Tumor Growth Inhibition Effect in MDA-MB-468 Xenograft Model

During the experimental period, there were no significant changes in the bodyweight of mice in the control group and in all treatment groups, suggesting that treatment with ruxolitinib and calcitriol as a single agent or in combination did not induce significant drug toxicity (Figure 5A). Although the single-agent treatment group of ruxolitinib or calcitriol did not show a significant decrease in tumor weight when compared to the control group, the tumor weight in the combination treatment group of ruxolitinib and calcitriol was significantly reduced compared to the control group and either single-agent treatment groups after 30 days of treatment (Figure 5B). Furthermore, the combination treatment with ruxolitinib and calcitriol induced a significant tumor volume reduction, especially at 21 days and after 26 days, unlike in the control group and single-agent treatment groups (Figure 5C,D). Western blot analysis showed that the ruxolitinib and calcitriol combination treatment group had significantly decreased expression levels of c-Myc and cyclin D1 and increased expression levels of c-PARP compared to the control and each single-agent treatment group (Figure 5E). Additionally, in the IHC staining results, the ruxolitinib and calcitriol combination treatment group showed significantly decreased expression levels of c-Myc and cyclin D1 and increased expression levels of c-PARP compared to the control group and each single-agent treatment group (Figure 5F). These results are consistent with the results obtained at the cellular level and verify the synergistic anticancer effect of ruxolitinib and calcitriol combination treatment in MDA-MB-468 cells, including cell cycle arrest and apoptosis.

## 3. Discussion

The combination of anticancer agents is the traditional treatment strategy for breast cancer [5]. In general, combination treatment potentially provides therapeutic advantages such as improved therapeutic efficacy, decreased toxicity, and reduced development of drug resistance compared with single-agent treatment [27,28]. In our previous reports, we demonstrated that combination treatment with ruxolitinib and calcitriol showed synergistic anticancer effects in luminal B subtype MCF7-HER18 breast cancer cells via increased apoptosis and altered expression of related proteins [25]. Owing to significant genomic, transcriptional, and biological heterogeneity in breast cancer [29], the response to treatment with various drug combinations may vary depending on the molecular subtype of breast cancer [30]. This reason could be an obstacle to applying our previous study results, which were limited to a specific breast cancer subtype. The maximum benefit from a therapeutic combination would be obtained following proper identification of the molecular subtypes of the disease, which would validate the use of specific anticancer agents that would provide the maximum therapeutic advantages [31]. Therefore, in the present study, we investigated the combined effects of ruxolitinib and calcitriol in other molecular subtypes of breast cancer cells, including the luminal A subtype MCF-7, HER2-enriched subtype SKBR3, and TN subtype MDA-MB-468, which were not covered in our previous study. We demonstrated the synergistic anticancer effect of ruxolitinib and calcitriol in SKBR3 and MDA-MB-468 cells via cell proliferation inhibition, cell cycle arrest, and increased apoptosis with alteration in related protein expression, including c-Myc, cyclin D1, CDKs, p21, p27, BAD, c-caspase, c-PARP, and p-p53. Furthermore, in the MDA-MB-468 tumor xenograft mouse model, we demonstrated the synergistic tumor growth inhibition effect of ruxolitinib and calcitriol combination treatment in a manner consistent with the findings of in vitro experiments, including the alteration of c-PARP, cyclin D1, and c-Myc expression without significant drug toxicity.

JAK2, a member of the JAK family of protein tyrosine kinases, is an important intracellular mediator of cytokine and hormone signaling, and a growing body of evidence demonstrates that the JAK2 signaling pathway is associated with the growth of solid tumor cells [32]. Previous studies reported that overexpression of JAK2 in breast cancer promotes cancer cell survival and growth by modulating cell proliferation and apoptosis, partly through the upregulation of cyclin D1 and Bcl-2 family members, such as Bcl-xL and Bcl-2 [33,34,35,36]. In clinical situations, JAK2 amplification in tumor samples is strongly associated with poor recurrence-free survival, poor overall survival, and low responsiveness to neoadjuvant chemotherapy in breast cancer patients [37,38]. Because of the crucial oncogenic role of the JAK2 pathway in breast cancer, several studies were conducted on the treatment of breast cancer by targeting JAK2 using JAK2 inhibitors. Ruxolitinib is a potent JAK1/2 inhibitor that was first approved for the treatment of myeloproliferative neoplasia (MPN) by the United States Food and Drug Administration (FDA) in 2011 and was subsequently used to investigate its anticancer effect in various solid tumors, including breast cancer [9,10,11,12]. To date, several studies reported significant anticancer effects of ruxolitinib as a single agent or in combination with other anticancer drugs in various molecular subtypes of breast cancer cells, in vitro and in vivo, mainly focusing on the TN subtype. Doheny et al. reported the cell proliferation inhibition effect of ruxolitinib in TN subtype MDA-MB-231, MDA-MB-468, and BT20 cells and reported the synergistic antitumor effect of ruxolitinib treatment combined with smoothened (SMO) inhibitors, including vismodegib and sonidegib, in a mouse xenograft model of TN subtype MDA-MB-231, MDA-MB-468, BT20 cells, and luminal B subtype BT474 cells [15]. Furthermore, other studies reported the in vitro and in vivo cell proliferation inhibition effect of ruxolitinib in HER2-enriched subtype SKBR3 and AU565 cells, and TN subtype HCC38 and SUM149 cells [39,40,41]. Similar to these reports, our study results show the significant anticancer effect of ruxolitinib in SKBR3 and MDA-MB-468 cells via the inhibition of cell proliferation, increased apoptosis, G2/M phase cell cycle arrest, and alteration of the expression of related proteins, including c-Myc, cyclin D1, CDKs, p21, c-PARP, p-p53, and c-caspase 3. Interestingly, the anticancer effect of ruxolitinib was relatively small in luminal A subtype MCF-7 cells compared with SKBR3 and MDA-MB-468 cells in our study results, which could be inferred from the higher IC_50_ values of MCF-7 cells (30.42 μM) than the IC_50_ values in other cell subtypes (13.94 μM in SKBR3 and 10.87 μM in MDA-MB-468 cells). In the same context, previous studies reported a low treatment effect of ruxolitinib in luminal A subtype breast cancer cells, including MCF-7 and T47D cells [12,42]. Although there are no studies reporting the different mechanisms of action of ruxolitinib treatment depending on various molecular subtypes of breast cancer, Escher et al. report that the treatment effect of ruxolitinib could be affected by the ER expression status of breast cancer cells [43]. In our study, MCF-7 cells treated with ruxolitinib showed different alterations in protein expression in comparison to other subtypes, including decreased expression of p27 and c-caspase 3. Taken together, these results suggest the different possible treatment effects of ruxolitinib depending on the molecular subtypes of breast cancer, which should be investigated in future studies.

Many studies reported the anticancer effect of calcitriol in various molecular subtypes of breast cancer cells. In a study by Jensen et al., calcitriol treatment inhibited cell proliferation, exhibited G0/G1 phase cell cycle arrest, and decreased expression of c-Myc protein in MCF-7 cells but was not associated with apoptosis [44]. Additionally, several previous studies reported the anticancer effect of calcitriol in TN subtype SUM-229PE and HER2-enriched subtype SKBR3 cells via cell proliferation inhibition and G0/G1 phase cell cycle arrest, but it was not associated with apoptosis [45,46]. Consistent with previous results, our study showed similar anticancer effects of calcitriol, with cell proliferation inhibition and G0/G1 phase cell cycle arrest with alterations in the expression of related proteins, including c-Myc, cyclins, CDKs, p21, and p27, but not by apoptosis in MCF-7, SKBR3, and MDA-MB-468 cells. Although the responsiveness to calcitriol treatment could be associated with not only the molecular subtype but also the vitamin D receptor (VDR) expression in breast cancer cells, considering that the VDR expression of breast cancer cell lines in our study was verified in previous studies [47,48]. Our study results suggest a consistent anticancer effect of calcitriol in various molecular subtypes of breast cancer, including luminal A, HER2-enriched, and TN subtypes.

The most important part of our study was that it showed the existence of different effects of the combination treatment with ruxolitinib and calcitriol among breast cancer cells of various molecular subtypes. Although ruxolitinib and calcitriol as single agents consistently exhibited significant anticancer effects in breast cancer cells with different molecular subtypes in our study, the combination of ruxolitinib and calcitriol showed a synergistic effect in HER2-enriched subtype SKBR3 and TN subtype MDA-MB-468 cells but did not show any synergistic treatment effect in luminal A subtype MCF-7 cells on cell proliferation and apoptosis analyses. Moreover, while the Western blot analysis results of SKBR3 and MDA-MB-468 cells mostly indicated the synergistic anticancer effect of ruxolitinib and calcitriol combination treatment on cell proliferation, cell cycle arrest, and apoptosis-related markers (decreased expression of c-Myc, cyclin D1, CDK1, CDK4, and increased expression of p21, p27, BAD, c-caspase 3, c-PARP, and p-p53), the result of MCF-7 cells indicated an antagonistic effect of ruxolitinib and calcitriol combination treatment on cell growth inhibition (decreased expression of p-p53 and increased expression of Bcl-2). This is an interesting finding because it raises the possibility of the existence of several different mechanisms of action that underlie the combination treatment effects of ruxolitinib and calcitriol depending on the molecular subtype of breast cancer. It is difficult to determine the different mechanisms, but we can speculate that the different treatment effects of ruxolitinib and calcitriol combination between the various molecular subtypes in our study may have been influenced by the crosstalk between JAK2 and ER-alpha (ERα). High ERα expression promotes the proliferation of ER-positive breast cancer cells [49], and one of the anticancer mechanisms of calcitriol in ER-positive breast cancer cells is the downregulation of ERα expression [50]. In a previous report by Gupta et al., JAK2 negatively regulated the transcriptional activity of ERα and its protein levels by enhancing its degradation, and silencing JAK2 in MCF-7 cells thereby resulting in increased ERα protein levels [51]. Based on these previous reports, it can be assumed that in our study, JAK2 suppression by ruxolitinib may have caused an increase in the expression of ERα in MCF-7 cells, which coincided with the effect of calcitriol in downregulating ERα expression and eventually resulted in an antagonistic effect of combined treatment with ruxolitinib and calcitriol in MCF-7 cells. In addition to ERα, JAK2 expression may be related to HER2 phosphorylation. Previously, Kavarthapu et al. reported that increased JAK2 expression might lead to HER2 phosphorylation and promote the proliferation of HER2-positive breast cancer cells through downstream MEK/ERK and PI3K/AKT pathways [52]. Based on these concepts, the synergistic treatment effect of ruxolitinib and calcitriol combination in luminal B subtype MCF7-HER18 breast cancer cells in our previous study could be interpreted as follows: the antagonistic effect of ruxolitinib and calcitriol combination treatment related to ERα expression can be compensated by the downregulation of HER2 phosphorylation through JAK2 suppression of ruxolitinib. Further investigations are warranted to determine the mechanisms of the different combination treatment effects of ruxolitinib and calcitriol in various molecular subtypes of breast cancer.

Despite the meaningfulness of this study’s findings, they should be interpreted with caution because of several limitations. First, although ruxolitinib inhibited JAK1 as well as JAK2 [40], we did not investigate JAK1 expression in this study. Second, because we did not include a normal, non-tumorigenic breast cell line in our study design, we could not investigate the effect of calcitriol and ruxolitinib on normal breast tissue. Third, there was a disparity between the PI3K mutation status of the breast cancer cell lines used in this study (MCF7: PI3K mutation-positive, SKBR3 and MDA-MB-468: PI3K mutation-negative), which could influence the therapeutic effect of calcitriol and ruxolitinib and thereby skew our study results. Lastly, although we evaluated the toxicity of ruxolitinib and calcitriol treatment based on the mouse body weight in our in vivo study, we could not rule out the potential hypercalcemic toxicity of calcitriol because we did not perform a blood test to check the calcium and phosphate levels in the tested mice.

## 4. Materials and Methods

### 4.1. Materials

Ruxolitinib was purchased from Selleckchem (S1378, Houston, TX, USA), and calcitriol was purchased from Sigma-Aldrich (D1530, St. Louis, MO, USA). Ruxolitinib and calcitriol were dissolved in dimethyl sulfoxide (DMSO) to prepare the stock solutions, and the final concentrations were obtained by diluting the stock solutions with RPMI-1640 medium (Gibco; Thermo Fisher Scientific Inc., Waltham, MA, USA). All solutions were prepared immediately prior to use. Antibodies against c-Myc (#13987), cyclin B1 (#12231), cyclin D1 (#2922), CDK1 (#9116), CDK4 (#12790), CDK6 (#13331), p21 (#2947), p27 (#3686), BAD (#9239), Bcl-2 (#15071), Bcl-xL (#2764), c-caspase 3 (#9661), PARP (#9532), c-PARP (#5625), p-p53 (#2521), p-JAK2 (#3776), and β-actin (#4967) were purchased from Cell Signaling Technology Inc. (Danvers, MA, USA).

### 4.2. Cells and Cell Culture

Human breast cancer cell lines MCF-7 (ER + PR + HER2 −: luminal A), SKBR3 (ER – PR − HER2 +: HER2-enriched), and MDA-MB-468 (ER − PR− HER2 −: TN) were purchased from the American Type Culture Collection (ATCC, Manassas, VA, USA). The cells were cultured in RPMI-1640 medium (Gibco; Thermo Fisher Scientific Inc., Waltham, MA, USA) supplemented with 10% fetal bovine serum (Corning Life Science B.V., Amsterdam, The Netherlands), 100 units/mL penicillin, and 100 μg/mL streptomycin. All the cells were cultured in a humidified incubator containing 5% CO_2_ at 37 °C.

### 4.3. Bromodeoxyuridine (BrdU) Assay of Cell Proliferation

Cell proliferation was quantified by measuring BrdU incorporation during DNA synthesis. This assay was performed according to the manufacturer’s protocol (Cell Proliferation ELISA BrdU, #11647229001; Roche Diagnostics GmbH, Mannheim, Germany). Briefly, MCF-7, SKBR3, and MDA-MB-468 cells (1 × 10^4^ cells/well) were separately seeded in triplicate into 96-well plates and allowed to grow overnight prior to treatment with various concentrations of ruxolitinib alone, calcitriol alone, or a combination of ruxolitinib and calcitriol for 72 h in a humidified incubator containing 5% CO_2_ at 37 °C. The cells were subsequently treated with the BrdU labeling solution for 2 h. The culture medium was removed, the cells were fixed using a fixative solution (3.7% formaldehyde in phosphate-buffered saline [PBS]) for 30 min at room temperature (RT), and the DNA was denatured. Cells were incubated with the anti-BrdU-POD solution for 90 min, and the unbound antibody conjugates were removed by washing the cells three times with 1X PBS. After washing, the tetra-methyl-benzidine substrate was added and incubated in the dark for 15 min at RT. Absorbance was quantified within 30 min at dual wavelengths of 450 nm and 540 nm using a microplate reader (VersaMax, Molecular Devices LLC, Sunnyvale, CA, USA).

### 4.4. Isobologram Analysis of the Interaction between Ruxolitinib and Calcitriol

The synergistic effects of ruxolitinib and calcitriol were examined using the Chou-Talalay combination index (CI) method [27,53]. The resultant CI values reflect the potential interactions between two drugs, where CI < 1 indicates a synergistic effect, CI = 1 indicates an additive effect, and CI > 1 indicates antagonism. The mean values from three independent experiments were used. Combination index analysis was performed using the Calcusyn software (ver. 2.0; Biosoft, Cambridge, UK).

### 4.5. Cell Cycle Analysis

All cell lines were seeded in 6-well plates (1 × 10^5^ cells/well). After incubation for 24 h in a humidified incubator containing 5% CO_2_ at 37 °C, the cells were treated with the anticancer agents, ruxolitinib alone, calcitriol alone, or a combination of ruxolitinib and calcitriol. A standard growth medium was used as the negative control. After 72 h, the cells were harvested, washed with PBS, and fixed in 70% ethanol at 4 °C for 24 h. Cells were then incubated with PBS containing 100 μg/mL RNase A and 100 μg/mL propidium iodide (PI) for 30 min at RT in the dark. Cell cycle analysis was performed using a Navios flow cytometer and the Kaluza software version 1.3 (Beckman Coulter Inc., Brea, CA, USA).

### 4.6. Apoptosis Assay

The apoptotic status of the cells was evaluated by flow cytometry using the Annexin V-fluorescein isothiocyanate (FITC) apoptosis detection kit (Southern Biotech, Birmingham, AL, USA) and PI staining, according to the manufacturer’s protocol. For all cell lines, a total of 1 × 10^6^ cells/mL were incubated with the anticancer agents, ruxolitinib alone, calcitriol alone, or a combination of ruxolitinib and calcitriol. A standard growth medium was used as the negative control. After 72 h, the cells were washed with PBS and suspended in a binding buffer containing 0.01 M HEPES, 140 mM NaCl and 2.5 mM CaCl_2_ at a final concentration of 1 × 10^6^ cells/mL. The cell suspension (100 μL containing 10^5^ cells) was incubated with 5 μL Annexin-FITC and 5 μL PI at RT for 15 min in the dark. Following this incubation, 400 μL of binding buffer was added, and the cells were analyzed by flow cytometry using a Navios flow cytometer with Kaluza software version 1.3 (Beckman Coulter Inc., Brea, CA, USA).

### 4.7. Western Blot Analysis

For Western blot analysis, cells were seeded in 6-well plates (1 × 10^6^ cells/well). Following incubation for 24 h in a humidified incubator containing 5% CO_2_ at 37 °C, cells were treated with the anticancer agent ruxolitinib alone, calcitriol alone, or a combination of ruxolitinib and calcitriol. A standard growth medium was used as the negative control. After 72 h, the cells were lysed using ionic detergent protein extraction buffer (PRO-PREP™, iNtrON Biotechnology, Suwon, Korea) containing phosphatase (1/100 mL lysis buffer, Sigma-Aldrich; Merck KGaA, Darmstadt, Germany) and protease inhibitors (1/100 mL lysis buffer, Sigma-Aldrich; Merck KGaA, St. Louis, MO, USA). Protein concentration was determined using the Bradford protein assay (Bio-Rad Laboratories, Hercules, CA, USA). A total of 20 μg of protein was resolved using 10% SDS-PAGE and transferred onto polyvinylidene fluoride membranes. The membranes were blocked for 1 h at RT with 5% skim milk in TBS-Tween-20. Membranes were probed overnight at 4 °C with the indicated 1:1000 diluted primary antibodies (all primary antibodies were obtained from Cell Signaling Technology Inc., Danvers, MA, USA), prior to washing and incubation with the anti-rabbit immunoglobulin G horseradish peroxidase-linked antibody (1:1000; #7074, Cell Signaling Technology, Inc., Danvers, MA, USA) for 1 h at RT. The membranes were developed, and protein signals were detected using enhanced chemiluminescence Western blotting detection reagents (GE Healthcare Life Sciences, Little Chalfont, UK). β-actin was used as a loading control in all Western blotting analyses, and the results were obtained by densitometry using MultiGauge software (ver. 2.0; FUJIFILM Corporation, Tokyo, Japan).

### 4.8. In Vivo Xenograft Model

Six-week-old male BALB/c nude mice (Orient Bio, Seongnam, Korea) were used in this study. Animal studies were approved by the Institutional Animal Care and Use Committee (IACUC) of the Catholic University of Korea (IRB No: IACUC-CUMC-2020-0129-02, 30 April 2020) and conducted in compliance with the guidelines of the Institute for Laboratory Animal Research, Korea. BALB/c nude mice were used for the comparative modeling of subcutaneous tumor growth. MDA-MB-468 cells (5 × 10^6^ cells) were subcutaneously injected into the right flank of each mouse. Mice were weighed three times per week. Five weeks after tumor cell injection, all mice had measurable tumors. Mice were then randomly grouped (*n* = 5 per group) and treated with normal saline (control), calcitriol (i.p., 0.4 µg/kg, three times a week at two- or three-day intervals), ruxolitinib (p.o., 45 mg/kg, three times a week at two or three days interval), and combining both agents (i.p. 0.4 µg/kg calcitriol combining with p.o. 45 mg/kg ruxolitinib, three times a week at two or three-day intervals) for 30 days. The stock of calcitriol and ruxolitinib was dissolved in DMSO followed by dilution with normal saline until a final DMSO concentration of 5% and 15%, respectively, was reached. After treatment completion, all mice were euthanized. Tumor size was measured twice weekly using calipers, and tumor volume was calculated using the formula length × width^2^ × 0.5236.

### 4.9. Immunohistochemical (IHC) Analyses

For immunohistochemical analysis, formalin-fixed paraffin-embedded (FFPE) tissue sections were deparaffinized and rehydrated using an ethanol series. The antigen was retrieved with 0.01 M citrate buffer (pH 6.0) by heating the sample in a microwave for 10 min. The tissue sections were then placed in 3% hydrogen peroxide for 5 min to inactivate endogenous peroxidases. Primary antibodies against c-PARP (1:100; Abcam), cyclin D1 (1:50; Abcam), and c-Myc (1:300; Novus Biologicals, Centennial, CO, USA) were used for IHC staining. Horseradish peroxidase (HRP)-conjugated secondary antibodies were obtained from Vector Laboratories. c-PARP, cyclin D1, and c-Myc levels were observed using a Pannoramic digital slide scanner system (3D HISTECH; Budapest, Hungary). Antibody validation was performed by serially diluting the antibodies in tissue sections on slides. The positive control experiment was performed using slides with tissue sections from cell-derived xenograft models, and the negative control experiment was performed by replacing the primary antibody with PBS.

### 4.10. Statistical Analysis

All data are expressed as the mean ± standard deviation, and all in vitro experiments were repeated at least three times. The normality of data distribution was determined using the Shapiro–Wilk test, and the difference between experimental groups was analyzed using the Mann–Whitney U test or Student’s *t*-test in accordance with data normality. For all tests, a *p*-value < 0.05 was considered statistically significant. Data were analyzed using SPSS (ver. 17.0; SPSS Inc., Chicago, IL, USA).

## 5. Conclusions

In summary, our study results showed a synergistic combination treatment effect of ruxolitinib and calcitriol in HER2-enriched subtype SKBR3 and TN subtype MDA-MB-468 breast cancer cells, but not in luminal A subtype MCF-7 cells in vitro. Furthermore, an in vivo study showed a synergistic tumor growth inhibition effect of ruxolitinib and calcitriol in the MDA-MB-468 tumor xenograft mouse model. To our knowledge, this is the first report to investigate the combined treatment effect of ruxolitinib and calcitriol in breast cancer cells of various molecular subtypes. Our results suggest that the combination treatment effect and mechanism of action of ruxolitinib and calcitriol combination treatment can differ depending on the molecular subtype of breast cancer. Our preclinical data provide evidence and support further investigations regarding combination treatment with ruxolitinib and calcitriol for breast cancer.

## Figures and Tables

**Figure 1 ijms-23-02535-f001:**
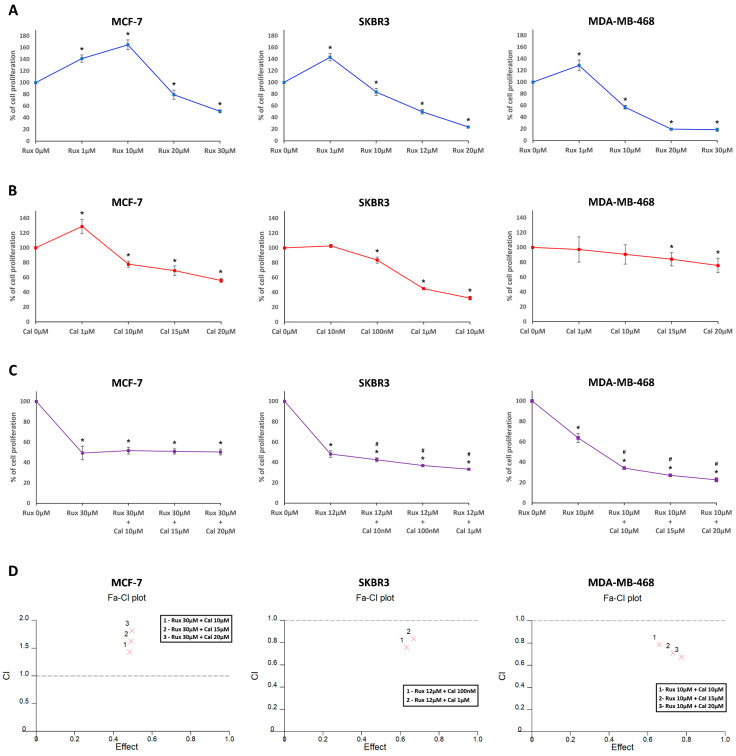
Combination effects of ruxolitinib and calcitriol in cell proliferation of MCF-7, SKBR3, and MDA-MB-468 breast cancer cells. Proliferation activity was determined by the BrdU assay after treatment with various concentrations of ruxolitinib and calcitriol in 72 h. (**A**) Ruxolitinib was treated as a single agent in MCF-7 (1 μM, 10 μM, 20 μM, and 30 μM), SKBR3 (1 μM, 10 μM, 12 μM, and 20 μM), and MDA-MB-468 (1 μM, 10 μM, 20 μM, and 30 μM) cells. (**B**) Calcitriol was treated as a single agent in MCF-7 (1 μM, 10 μM, 15 μM, and 20 μM), SKBR3 (10 nM, 100 nM, 1 μM, and 10 μM), and MDA-MB-468 (1 μM, 10 μM, 15 μM, and 20 μM) cells. (**C**) Ruxolitinib and calcitriol were treated as combination in MCF-7 (ruxolitinib 30 μM and calcitriol 10 μM, ruxolitinib 30 μM and calcitriol 15 μM, ruxolitinib 30 μM and calcitriol 20 μM), SKBR3 (ruxolitinib 12 μM and calcitriol 10 nM, ruxolitinib 12 μM and calcitriol 100 nM, ruxolitinib 12 μM and calcitriol 1 μM), and MDA-MB-468 (ruxolitinib 10 μM and calcitriol 1 μM, ruxolitinib 10 μM and calcitriol 10 μM, ruxolitinib 10 μM and calcitriol 20 μM) cells. (**D**) The CI versus fraction affected plot was calculated using Calcusyn software and represents the drug combination effects. Synergy was defined as CI value < 1.0, antagonism as CI value > 1.0, and additivity as a CI value = 1.0. * *p* < 0.05, compared with control, # *p* < 0.05, compared with ruxolitinib single-agent treatment. The data presented are the average of three experiments, each represented as the mean ± SD. Rux, ruxolitinib; Cal, calcitriol; CI, combination index; Fa, fraction affected.

**Figure 2 ijms-23-02535-f002:**
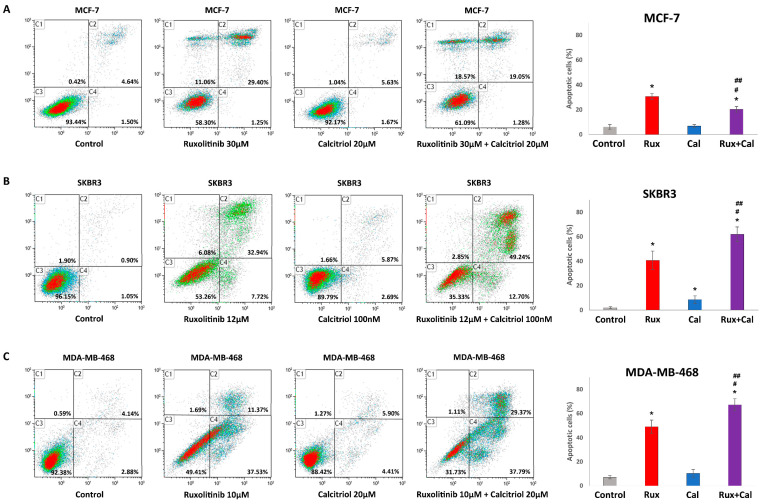
Combination effects of ruxolitinib and calcitriol in apoptosis of MCF-7, SKBR3, and MDA-MB-468 breast cancer cells. Apoptosis profile was assessed by Annexin V/PI assay by flow cytometry after treatment with various concentrations of ruxolitinib and calcitriol in 72 h. (**A**) MCF-7 cells were treated with ruxolitinib 30 μM, calcitriol 20 μM, or ruxolitinib 30 μM and calcitriol 20 μM combination. (**B**) SKBR3 cells were treated with ruxolitinib 12 μM, calcitriol 100 nM, or ruxolitinib 12 μM and calcitriol 100 nM combination (**C**) MDA-MB-468 cells were treated with ruxolitinib 10 μM, calcitriol 20 μM, or ruxolitinib 10 μM and calcitriol 20 μM combination. * *p* < 0.05 compared with control, # *p* < 0.05 compared with ruxolitinib single-agent treatment, ## *p* < 0.05 compared with calcitriol single-agent treatment. The data presented are the average of three experiments, each represented as the mean ± SD. Rux, ruxolitinib; Cal, calcitriol.

**Figure 3 ijms-23-02535-f003:**
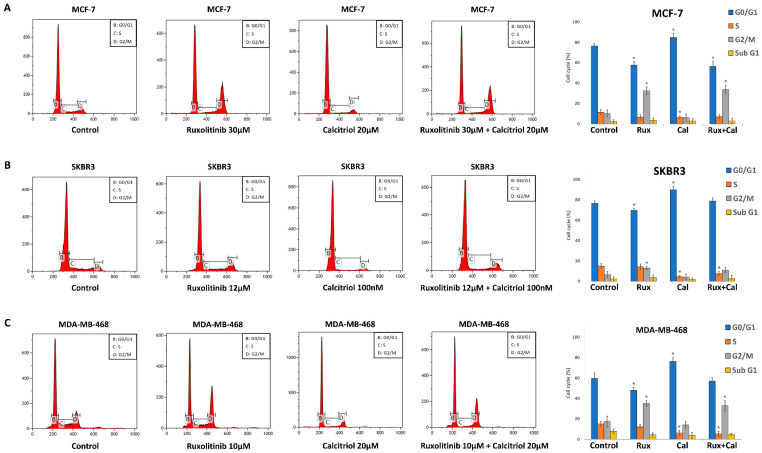
Combination treatment effects of ruxolitinib and calcitriol in cell cycle regulation of MCF-7, SKBR3, and MDA-MB-468 breast cancer cells by flow cytometry after treatment with various concentrations of ruxolitinib and calcitriol in 72 h. (**A**) MCF-7 cells were treated with ruxolitinib 30 μM, calcitriol 20 μM, or ruxolitinib 30 μM and calcitriol 20 μM combination (**B**) SKBR3 cells were treated with ruxolitinib 12 μM, calcitriol 100 nM, or ruxolitinib 12 μM and calcitriol 100 nM combination (**C**) MDA-MB-468 cells were treated with ruxolitinib 10 μM, calcitriol 20 μM, or ruxolitinib 10 μM and calcitriol 20 μM combination. The histogram present the proportions of cells in each phase of cell cycles. * *p* < 0.05 compared with control. The data presented are the average of three experiments, each represented as the mean ± SD. Rux, ruxolitinib; Cal, calcitriol.

**Figure 4 ijms-23-02535-f004:**
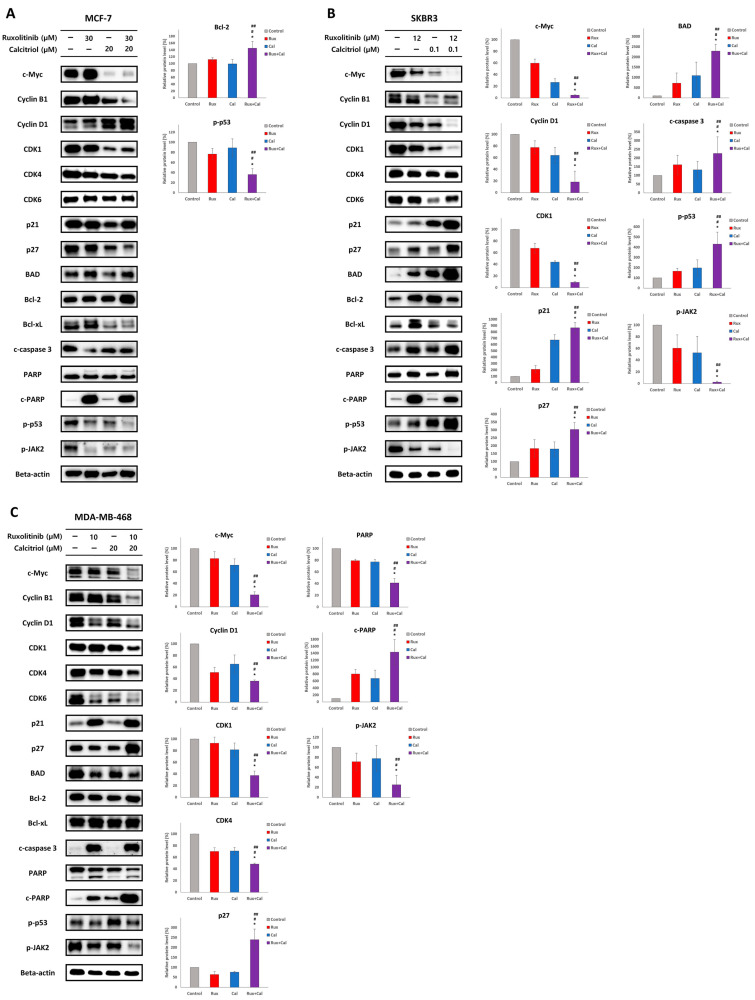
Combination treatment effects of ruxolitinib and calcitriol on the expression of cell signaling proteins in MCF-7, SKBR3, and MDA-MB-468 breast cancer cells by Western blot analysis after treatment with various concentrations of ruxolitinib and calcitriol in 72 h. (**A**) In MCF-7 cells, combination treatment with ruxolitinib 30 μM and calcitriol 20 μM significantly decreased p-p53 expression and increased Bcl-2 expression in comparison to that with control, ruxolitinib 30 μM treatment, and calcitriol 20 μM treatments. (**B**) In SKBR3 cells, combination treatment with ruxolitinib 12 μM and calcitriol 100 nM significantly decreased c-Myc, cyclin D1, CDK1, and p-JAK2 expression and increased p21, p27, BAD, c-caspase 3, p-p53 expression in comparison to that with control, ruxolitinib 12 μM treatment, and calcitriol 100 nM treatments. (**C**) In MDA-MB-468 cells, combination treatment with ruxolitinib 10 μM and calcitriol 20 μM significantly decreased c-Myc, cyclin D1, CDK1, CDK4, and p-JAK2 expression and increased p27 and c-PARP expression in comparison to that with control, ruxolitinib 10 μM treatment, and calcitriol 20 μM treatments. The data presented are the average of three experiments, each represented as the mean ± SD. * *p* < 0.05 compared with control, # *p* < 0.05 compared with ruxolitinib single-agent treatment, ## *p* < 0.05 compared with calcitriol single-agent treatment. Rux, ruxolitinib; Cal, calcitriol.

**Figure 5 ijms-23-02535-f005:**
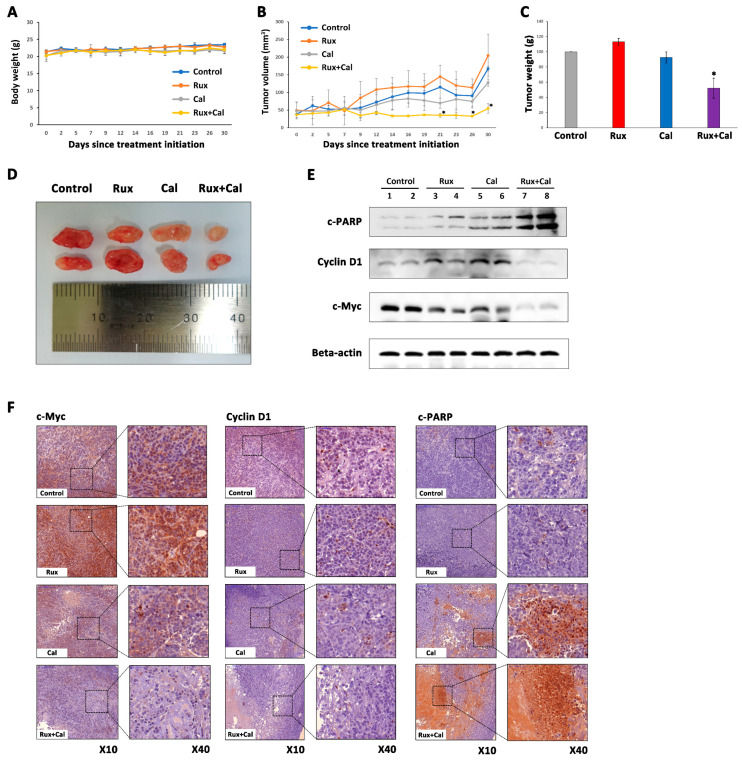
Combination anticancer effects of ruxolitinib and calcitriol in MDA-MB-468 breast cancer cells xenografts. Six-week-old male BALB/c mice were treated with normal saline (control), calcitriol (0.4 µg/kg, three times a week), ruxolitinib (45 mg/kg, three times a week), and both agents (0.4 µg/kg calcitriol and 45 mg/kg ruxolitinib, three times a week) for 30 days. (**A**) The body weight was measured three times a week. (**B**) Tumor weight was measured 30 days after initiation of treatment. (**C**) Tumor volume was measured three times a week until 30 days after treatment initiation. (**D**) Mice bearing tumors were sacrificed 30 days after treatment initiation, and the tumors are shown. (**E**) The protein expression levels of c-PARP, cyclin D1, and c-Myc in tumor tissues were analyzed by Western blotting. (**F**) Formalin-fixed paraffin-embedded tumor sections were stained with H&E and c-Myc, cyclin D1, and c-PARP. The data are represented as the mean ± SD. * *p* < 0.05 compared with control, ruxolitinib single-agent treatment and calcitriol single-agent treatment. Rux, ruxolitinib; Cal, calcitriol.

**Table 1 ijms-23-02535-t001:** Combination index for various concentrations of ruxolitinib and calcitriol combination treatment in MCF-7, SKBR3, and MDA-MB-468 cells.

	Ruxolitinib 30 µM			Ruxolitinib 12 µM		Ruxolitinib 10 µM		
	Calcitriol 10 µM	Calcitriol 15 µM	Calcitriol 20 µM	Calcitriol 100 nM	Calcitriol 1 µM	Calcitriol 10 µM	Calcitriol 15 µM	Calcitriol 20 µM
MCF-7	1.433	1.625	1.814	-	-	-	-	-
SKBR3	-	-	-	0.759	0.836	-	-	-
MDA-MB-468	-	-	-	-	-	0.787	0.711	0.676

## Data Availability

The data presented in this study are available on request from the corresponding author.

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
