# Peer review of "Effects of Ruxolitinib and Calcitriol Combination Treatment on Various Molecular Subtypes of Breast Cancer"

_ijms, 2022, doi:10.3390/ijms23052535_

Round 1

Reviewer 1 Report

The manuscript contains a lot of interesting results. In vivo studies are the manuscript's strengths, but I have a few comments:

  1. In this study were used tree human breast cancer cell lines representing different sybtype of breast cancer, but there is no normal, non-tumorigenic cell line.
  2. The solvent used for calcitriol and ruxolitinib was not specified in the study. Calcitriol is soluble in organic solvents such as ethanol, methanol, and DMSO. Ruxolitinib is soluble in DMSO and ethanol. Both calcitriol and ruxolitinib are insoluble in water. In the manuscript is: „The final concentrations were obtained by diluting the stock solutions with RPMI-1640 medium.” But the authors did not mention how the stock solutions were prepared.

Author Response

Point 1: In this study were used tree human breast cancer cell lines representing different subtype of breast cancer, but there is no normal, non-tumorigenic cell line.

Response 1)

- Thank you for your observant comment. Unfortunately, we did not investigate the therapeutic effect of ruxolitinib and calcitriol in normal, non-tumorigenic breast cell lines in this study, because we aimed to investigate the anticancer effect of ruxolitinib and calcitriol only in breast cancer cells. We agree with you that the lack of experimentation using a normal, non-tumorigenic cell line was a drawback of our study. Therefore, we have noted this fact as one of the limitations of our study in the discussion section.

Point 2: The solvent used for calcitriol and ruxolitinib was not specified in the study. Calcitriol is soluble in organic solvents such as ethanol, methanol, and DMSO. Ruxolitinib is soluble in DMSO and ethanol. Both calcitriol and ruxolitinib are insoluble in water. In the manuscript is: „The final concentrations were obtained by diluting the stock solutions with RPMI-1640 medium.” But the authors did not mention how the stock solutions were prepared.

Response 2)

- Thank you for your valuable comment. In consideration of your comment, we mentioned that “Ruxolitinib and calcitriol were dissolved in dimethyl sulfoxide (DMSO) to prepare the stock solutions,” in the revised 4.1 Materials section.

- Additionally, we are sorry to say that the reference numbers were incorrectly written in the original manuscript. Therefore, we have corrected the reference numbers in the revised version.

- We have also revised the author information as it was changed from that written in the initial manuscript.

Reviewer 2 Report

 In the presented paper Lim et al. tested the combination of the active form of vitamin D3
calcitriol and ruxolitinib
in vitro and in vivo. Three different types of breast cancer cells lines
were used MCF-7, SKBR3, and MDA-MB-468. Results showed that the combination of
calcitriol and ruxolitinib has a synergistic effect in SKBR3 and MDA-MB-468.
The advantages of the prepared article are:
- the article is written in a thoughtful and orderly way, the data is presented in a logical
and understandable way
- very well-conducted studies of inhibition of the proliferation of single substances and
selection of combination doses based on the IC
50 value
- the presented western blot results are in good and legible quality
After reading the article, there are a few substantive questions and issues to be clarified:
In vitro
1. Why cell lines used for proliferation assay were incubated with drugs for 72h at room
temperature? What was the viability of cells after 72h incubation at room
temperature? How this could impact the results in comparison to the cells after
incubation in humidified CO
2 incubator?
2. In the Western blot panel there are several proteins presented including JAK2, one of
the ruxolitinib targets. Have the authors also tested the level of JAK1 protein, which
is also one of the molecular targets of ruxolitinib? If the level of this protein has not
been tested, please explain why.
3. The authors indicate that only in the case of the MCF-7 line, no synergism was
observed. They emphasize that this effect is due to molecular differences. The MCF-
7 lineage has a PI3K mutation, which may be one of the reasons for resistance to the
action of ruxolitinib. There is no mention of this mutation in the MCF-7 lineage in the
discussion, it would be good to mention it.
In vivo
4. Please describe in the material and methods how the calcitriol and ruxolitinib were
prepared for mice. How was it dissolved? If several dissolvents were used, what were
their final concentrations?
5. Authors present IHC results for c-Myc, Cyclin D1, c-PARP. How were the antibodies
used for staining validated? Were the appropriate positive and negative controls
performed to rule out the possibility of non-specific staining?.
6. Calcitriol is known for its high calcemic activity. The authors emphasized that no
toxicity was observed during
in vivo studies based on the observation of body weight.
Do the authors know whether the concentration of calcitriol used in vivo causes
calcemic toxicity? Were there any tests on the level of calcium ions and phosphorates
in the blood of the tested mice?
Spell check required:
- In vitro and in vivo as Latin words, should be italicized:
in vivo, in vitro
- vitamin D3 and IC50 should be written using subscript: IC50, vitamin D3
- all text should be justified (distribute evenly between the margins)
- in the text there are several words with a dash in the middle, like: path-ways, de-cresed.

Author Response

Reviewer 2

Comments and Suggestions for Authors

 In the presented paper Lim et al. tested the combination of the active form of vitamin D3 – calcitriol and ruxolitinib in vitro and in vivo. Three different types of breast cancer cells lines were used MCF-7, SKBR3, and MDA-MB-468. Results showed that the combination of calcitriol and ruxolitinib has a synergistic effect in SKBR3 and MDA-MB-468.

The advantages of the prepared article are:
- the article is written in a thoughtful and orderly way, the data is presented in a logical and understandable way
- very well-conducted studies of inhibition of the proliferation of single substances and selection of combination doses based on the IC50 value
- the presented western blot results are in good and legible quality

After reading the article, there are a few substantive questions and issues to be clarified:

In vitro

Point 1: Why cell lines used for proliferation assay were incubated with drugs for 72h at room temperature? What was the viability of cells after 72h incubation at room temperature? How this could impact the results in comparison to the cells after incubation in humidified CO2 incubator?

Response 1)

- Thank you for your insightful comment. We are sorry to say that “at room temperature” was incorrectly written. We have revised this portion to “in a humidified incubator containing 5% CO2 at 37 °C.”

- Although not reported in our manuscript, the therapeutic effect of drugs in all cell lines was increased in a time-dependent manner (24 h < 48 h < 72 h) according to the results of our preliminary study. Therefore, we decided to consider the treatment duration as 72 h. If you recommend it, we will gladly mention these results in our manuscript.

Point 2: In the Western blot panel there are several proteins presented including JAK2, one of the ruxolitinib targets. Have the authors also tested the level of JAK1 protein, which is also one of the molecular targets of ruxolitinib? If the level of this protein has not been tested, please explain why.

Response2)

- Thank you for your valuable comment. Unfortunately, we did not investigate the therapeutic effect of ruxolitinib and calcitriol on JAK1 expression because we focused only on their effect on JAK2 expression in this study. However, we agree with you that because ruxolitinib inhibits both JAK1 and JAK2 significantly, the absence of any experiment on JAK1 expression is a drawback of our study. Therefore, we noted this as one of the limitations of our study in the discussion section.

Point 3: The authors indicate that only in the case of the MCF-7 line, no synergism was observed. They emphasize that this effect is due to molecular differences. The MCF-7 lineage has a PI3K mutation, which may be one of the reasons for resistance to the action of ruxolitinib. There is no mention of this mutation in the MCF-7 lineage in the discussion, it would be good to mention it.

Response 3)

- Thank you for your insightful comment. As you said, there was a disparity between the PI3K mutation status of the breast cancer cell lines used in our study, which could influence the therapeutic effect of calcitriol and ruxolitinib. As we did not consider this fact during the study, we have noted this disparity as one of the limitations of our study in the discussion section.

In vivo

Point 4: Please describe in the material and methods how the calcitriol and ruxolitinib were prepared for mice. How was it dissolved? If several dissolvents were used, what were their final concentrations?

Response 4)

- Thank you for your thoughtful comment. We described the dissolution of calcitriol and ruxolitinib in subsection 4.8 of the Materials and Methods section as follows: “The stock solutions of calcitriol and ruxolitinib were dissolved in DMSO, followed by dilution with normal saline until a final DMSO concentration of 5% and 15%, respectively, was reached.”

Point 5: Authors present IHC results for c-Myc, Cyclin D1, c-PARP. How were the antibodies used for staining validated? Were the appropriate positive and negative controls performed to rule out the possibility of non-specific staining?

Response 5)

Thank you for your valuable comment. As suggested, we have described the procedures followed for antibody validation, positive control, and negative control in subsection 4.9 of the Materials and methods section as follows: “Antibody validation was performed by serially diluting the antibodies in tissue sections on slides. The positive control experiment was performed using slides with tissue sections from cell-derived xenograft models, and the negative control experiment was performed by replacing the primary antibody with PBS.” Because the therapeutic effect of ruxolitinib combined with calcitriol on c-Myc, cyclin d1, and c-PARP expression was demonstrated in our in vitro study, we used slides of tissue sections obtained from cell-derived xenograft models as the positive control.

Point 6: Calcitriol is known for its high calcemic activity. The authors emphasized that no toxicity was observed during in vivo studies based on the observation of body weight. Do the authors know whether the concentration of calcitriol used in vivo causes calcemic toxicity? Were there any tests on the level of calcium ions and phosphorates in the blood of the tested mice?

Response 6)

- Thank you for your valuable comment. Unfortunately, we did not investigate the serum level of calcium and phosphate in the tested mice. We agree with you that there was a possibility of hypercalcemic toxicity which could not be judged by evaluating weight change. Therefore, we noted this fact as one of the limitations of our study in the discussion section.

Spell check required:

- In vitro and in vivo as Latin words, should be italicized: in vivo, in vitro

Response)

- Thank you for your comment. We have italicized in vitro and in vivo throughout the manuscript.

- vitamin D3 and IC50 should be written using subscript: IC50, vitamin D3

Response)

- Thank you for your comment. We have subscripted the numbers in IC50 and vitamin D3.

- all text should be justified (distribute evenly between the margins)

Response)

- Thank you for your comment. We have justified all the text as suggested.

- in the text there are several words with a dash in the middle, like: path-ways, de-cresed.

Response)

- Thank you for your comment. We have revised the relevant words appropriately. (pro-vide, path-ways, sub-types, in-creased, de-creased, rep-resented, down-regulated, in-crease, com-pared, pre-formed)

- Additionally, we are sorry to say that the reference numbers were incorrectly written in the original manuscript. Therefore, we have corrected the reference numbers in the revised version.

- We have also revised the author information as it was changed from that written in the initial manuscript.